

# The importance of spawning behavior in understanding the vulnerability of exploited marine fishes in the U.S. Gulf of Mexico

Christopher R. Biggs[1], William D. Heyman[2], Nicholas A. Farmer[3], Shin'ichi Kobara[4], Derek G. Bolser[1,8], Jan Robinson[5], Susan K. Lowerre-Barbieri[6] and Brad E. Erisman[1,7]

[1] Marine Science Institute, The University of Texas at Austin, Port Aransas, Texas, United States
[2] LGL Ecological Research Associates, Inc., Bryan, Texas, United States
[3] Southeast Regional Office, NOAA National Marine Fisheries Service, St. Petersburg, Florida, United States
[4] Department of Oceanography, Texas A&M University, College Station, Texas, United States
[5] Australian Research Council (ARC) Centre of Excellence for Coral Reef Studies, James Cook University, Townsville, Queensland, Australia
[6] Fisheries and Aquatic Science Program, School of Forest Resources and Conservation, University of Florida, Gainesville, Florida, United States
[7] Current Affiliation: Fisheries Resources Division, Southwest Fisheries Science Center, National Marine Fisheries Service, National Oceanic and Atmospheric Administration, La Jolla, California, United States
[8] Current Affiliation: Cooperative Institute for Marine Resources Studies, Hatfield Marine Science Center, Oregon State University, Newport, Oregon, United States

Corresponding author
Brad E. Erisman,
brad.erisman@noaa.gov

## ABSTRACT

The vulnerability of a fish stock to becoming overfished is dependent upon biological traits that influence productivity and external factors that determine susceptibility or exposure to fishing effort. While a suite of life history traits are traditionally incorporated into management efforts due to their direct association with vulnerability to overfishing, spawning behavioral traits are seldom considered. We synthesized the existing biological and fisheries information of 28 fish stocks in the U.S. Gulf of Mexico to investigate relationships between life history traits, spawning behavioral traits, management regulations, and vulnerability to fishing during the spawning season. Our results showed that spawning behavioral traits were not correlated with life history traits but improved identification of species that have been historically overfished. Species varied widely in their intrinsic vulnerability to fishing during spawning in association with a broad range of behavioral strategies. Extrinsic vulnerability was high for nearly all species due to exposure to fishing during the spawning season and few management measures in place to protect spawning fish. Similarly, several species with the highest vulnerability scores were historically overfished in association with spawning aggregations. The most vulnerable species included several stocks that have not been assessed and should be prioritized for further research and monitoring. Collectively, the results of this study illustrate that spawning behavior is a distinct aspect of fish ecology that is important to consider for predictions of vulnerability and resilience to fisheries exploitation.

# INTRODUCTION

The vulnerability of a stock, population, or species of marine fish to become overfished or experience overfishing is dependent upon both intrinsic aspects of its evolutionary history, ecology, and population biology as well as extrinsic factors related to the fishery and its management that determine the level of exposure to fishing pressure (*Jennings, Reynolds & Mills, 1998*; *Dulvy et al., 2004*; *Patrick et al., 2010*). Intrinsic vulnerability is a function of various life history traits (e.g., growth rate or longevity) and behavioral traits (e.g., spatiotemporal spawning patterns) that influence stock productivity and resilience: the capacity of a fish population to recover once it becomes depleted (*Adams, 1980*; *Reynolds, Jennings & Dulvy, 2001*; *Stobutzki, Miller & Brewer, 2001*). Conversely, extrinsic vulnerability factors are linked to the dynamics of the fishery (e.g., fishing effort or catch efficiency), the effectiveness of management policies, and governance structure that combine to determine stock susceptibility and the potential for the fishery to negatively impact stock condition (*Cinner et al., 2009*; *Hobday et al., 2011*; *Leslie et al., 2015*). In situations where insufficient information exists to perform quantitative assessments of biomass or modelling of population dynamics, intrinsic and extrinsic attributes associated with productivity of the stock and exposure to fishing pressure can be used to estimate the overall vulnerability of a stock relative to others in the same region (*Frisk, Miller & Dulvy, 2005*; *Patrick et al., 2010*; *Hobday et al., 2011*). Furthermore, vulnerability analyses and conservation evaluations have been useful in identifying stocks that should be prioritized for additional research and monitoring (*Morato, Cheung & Pitcher, 2006*; *Davies & Baum, 2012*; *Mamauag et al., 2013*).

A suite of life history traits is associated with a high intrinsic vulnerability to becoming overfished. Fish species that are slow growing, long-lived, late to mature, and experience low natural mortality are consistently linked to reduced resilience and increased risk of population collapse in response to fishing (*Jennings, Reynolds & Mills, 1998*; *Musick, 1999*; *King & McFarlane, 2003*; *Winemiller, 2005*). Moreover, certain life history traits correlate with each other as intrinsic indicators of vulnerability or compensatory capacity (*Dulvy et al., 2004*; *Kindsvater et al., 2016*). For example, fish with slow growth rates tend to have low natural mortalities and late onset of sexual maturity, although there are exceptions (see *Coulson, 2019*). Life history traits (e.g., growth, reproduction, and death rates) are integral to data-limited stock assessments, but they are also used within data-rich stock assessments (e.g., length or age-structured models) when sufficient data are available (*Hilborn & Walters, 1992*; *Methot & Wetzel, 2013*). Likewise, most vulnerability assessments are designed to account for vulnerability associated with life history traits. However, certain types of spawning behaviors and reproductive patterns are also associated with a high intrinsic vulnerability to fishing but are not typically incorporated into assessment frameworks and thus do not account for this source of vulnerability.

Spawning behavior is associated with productivity (*Cheung, Pitcher & Pauly, 2005*) and resilience (*Lowerre-Barbieri et al., 2017*) in marine fishes such that the spatial and temporal components of spawning may affect the relationship between stocks and recruitment, affecting how a species responds to fishing pressure (*Maunder & Deriso, 2013*; *Donahue et al., 2015*; *Erisman et al., 2017a*). For instance, the number of spawning sites and number of spawning opportunities are positively correlated with increased reproductive resilience (*Erisman et al., 2011*; *Lowerre-Barbieri et al., 2015*). The duration of the spawning season is inversely related to vulnerability, in which species with predictable but brief spawning periods are associated with the most rapid and severe population declines compared to those that spawn year-round or over protracted seasons (*Mullon, Fréon & Cury, 2005*; *Claro et al., 2009*; *Sadovy De Mitcheson & Erisman, 2012*). Large changes in fish densities or relative abundance in association with spawning are directly linked to marked increases in catchability, which also increases susceptibility and overall vulnerability to fishing (*Wilberg et al., 2009*; *Erisman et al., 2011*, *2014*; *Robinson & Samoilys, 2013*; *Robinson, 2015*). Catchability is an important factor in fisheries assessments (*Hilborn & Walters, 1992*; *Arreguín-Sánchez, 1996*) but can be difficult to estimate, as it is affected by a combination of extrinsic factors (fishery and management related) and intrinsic factors such as aggregating behavior and changes in relative abundance (*Skjold, Eide & Flaaten, 1996*; *Solmundsson, Karlsson & Palsson, 2003*; *Erisman et al., 2011*).

The continuum of adult behavioral dynamics that collectively determine the spatial and temporal characteristics of spawning can be described based on the degree of aggregating behavior, the duration of spawning season, and the change in relative abundance of fish during spawning (*Claydon, Mccormick & Jones, 2014*; *Lowerre-Barbieri et al., 2017*; *Erisman et al., 2017b*). On one end of the spectrum are transient aggregations, which include individuals that have migrated from within a large catchment area to congregate in high densities at very specific locations during predictable periods (*Domeier, 2012*). On the other end of the spectrum are species that do not aggregate to spawn or exhibit simple migratory behavior, in which an entire group or population moves from a foraging ground to a spawning area without a change in relative abundance (*Domeier, 2012*). Also, within the spectrum of spawning behaviors are resident aggregations that include fishes that form small spawning aggregations of a few to a few dozen individuals, which often occur throughout the year and draw from small catchment areas. Lastly, some populations or species of fish are mixed spawners that employ a combination of resident and transient spawning behaviors in different areas and times (*Lowerre-Barbieri et al., 2009*; *Tinhan et al., 2014*).

Understanding interspecific variations in reproductive behaviors including reproductive migration patterns, changes in relative abundance, and the timing, duration, and spatial distribution of spawning activities may help scientists and managers better understand the intrinsic vulnerability of a species to fishing in relation to spawning and manage for resilience. Yet, the reproductive potential of a stock is typically measured based on spawning stock biomass or total egg production (*i.e.*, fecundity), rather than traits affecting reproductive resilience (*Lowerre-Barbieri et al., 2017*), including spatiotemporal behavioral traits, although spawning season duration is used to estimate annual fecundity
in indeterminate species (*Cooper et al., 2013*; *Maunder & Deriso, 2013*; *Ganias, Somarakis & Nunes, 2014*). While spawning behavior traits have been considered in some vulnerability analyses (*Cheung, Pitcher & Pauly, 2005*; *Erisman et al., 2014*; *Robinson & Samoilys, 2013*; *Robinson et al., 2015*), this aspect of fish ecology remains poorly studied, under-utilized for assessing stock health, and rarely emphasized in management frameworks (e.g., *Erisman et al., 2011*; *Sadovy De Mitcheson & Erisman, 2012*; *Cheung et al., 2013*).

Given the influence life history and spawning behavioral traits have on vulnerability, it is important to consider whether these traits are correlated and how well they explain vulnerability to fishing during spawning and exploitation status. If life history and spawning behavior traits are not correlated, then for situations in which spawning behavior improves predictions of vulnerability, increased research efforts to understand spawning behaviors would help identify vulnerable species that may otherwise be overlooked and identify areas where targeted protection of spawning fish may be needed to maintain sustainable harvest levels or rebuild overfished stocks (*Grüss & Robinson, 2014*; *Grüss et al., 2018*). Studies that have examined the relationship between life history traits and spawning behavior were conducted in tropical regions and did not focus explicitly on exploited species (e.g., *Choat, 2012*; *Nemeth, 2012*). Therefore, uncovering this relationship and identifying species whose vulnerability to fishing is not explained by life history traits alone would assist managers in prioritizing research, monitoring, and assessment efforts accordingly.

The U.S. Gulf of Mexico (GOM) presents an ideal opportunity to answer important questions about relationships between life history traits, reproductive behavior, and the vulnerability of exploited marine fishes during the spawning season. The GOM is a highly productive system that supports a diverse set of taxa (*i.e.*, numerous families) of highly exploited fish species that exhibit a wide range of life history strategies and reproductive patterns (*Farmer et al., 2016*; *Biggs et al., 2017*). There is extensive information on life history characteristics for most managed species, and the majority of fisheries in state and federal waters rely heavily upon life history data as the basis for assessments of both data-limited and data-rich stocks (*Sagarese et al., 2015*; *SEDAR, 2016a*). There is also a growing recognition of the need to incorporate reproductive behavior in the conservation and management of these and other exploited fishes in the region (*Lowerre-Barbieri, Burnsed & Bickford, 2016*; NOAA RESTORE Science Program, *Kobara et al., 2017*; *Grüss et al., 2018*; *Erisman et al., 2018*; *Heyman et al., 2019*). Therefore, it is a good system to compare exploitation history to spawning behavior and evaluate the degree to which more attention to spawning patterns in relation to fishing vulnerability is warranted.

The objective of this study was to investigate relationships between life history and spawning behavioral traits and identify patterns among exploited species in the GOM that have been historically overfished or may be particularly vulnerable to exploitation during the spawning season. We employed multiple methods to evaluate these relationships. First, we used a data synthesis approach to test whether life history and spawning behavioral traits were correlated. Second, we used ordination to identify groups of traits (life history and spawning behavior) that were common among overfished species. Third, we

conducted a vulnerability analysis to identify species and stocks that are likely highly vulnerable during the spawning season and should be prioritized for further research and monitoring. The results indicate whether including spawning behavior characteristics can improve our ability to assess the vulnerability and resilience of marine fishes to exploitation. Additionally, this information will identify important data gaps in our understanding of the spawning behavior of exploited marine fishes and provide the basis for further research on interactions between spawning behavior and fishing activities. This information will be applicable to the management and monitoring of exploited marine fishes in the GOM, and the approach should be transferable to regional vulnerability assessments of fish stocks elsewhere.

## MATERIALS & METHODS

### Species selection

A hierarchical ranking process was used to identify a manageable number of relevant species to include in the analysis (*Biggs et al., 2017*). The preliminary list of species included commonly occurring and commonly caught recreational or commercial species that inhabit either offshore, coastal or estuarine waters of the GOM, including all species managed in United States federal waters by the Gulf of Mexico Fisheries Management Council (GMFMC). Species were scored based on their aggregating behavior associated with spawning and a fisheries index, which included two aspects of management status (inclusion in GMFMC's fisheries management plan and NOAA's fish stock sustainability index; https://www.fisheries.noaa.gov), importance to commercial fisheries (based on total annual landings in kg.), importance to recreational fisheries (based on total annual landing in number of fish), and their endangered status according to the IUCN Red List (https://www.iucnredlist.org). A detailed description of the selection process is available in Material S1.

### Life history and spawning behavioral traits

We compiled information on the life history and spawning behavior for the selected species through reviews of primary literature, technical reports, and stock assessments from NOAA's Southeast Data Assessment and Review (SEDAR; http://sedarweb.org/). Life history parameters included maximum age ($A_{max}$), maximum weight ($W_{max}$) and maximum length ($L_{max}$), von Bertalanffy growth coefficient ($k$), asymptotic length ($L_{inf}$), age ($A_m$) and length at maturity ($L_m$), and rate of natural mortality ($M$) (Table 1). These parameters were chosen, because they have been shown to be directly associated with vulnerability to fishing pressure, and they are commonly used in productivity-susceptibility analyses, stock assessments, and in defining species stock complexes (*Patrick et al., 2009*; *Robinson, 2015*; *Farmer et al., 2016*). The reported values were specific to the GOM unless there were no data, in which case information from the Atlantic or Caribbean was used. When multiple values were found, the average (±SE) was used. Sexual pattern was not included in this study, because specific traits associated with sexual pattern (e.g., diagnosis, sex ratios, timing of sex change) are unknown for most hermaphroditic species in the GOM. Variations in such traits strongly influence the resilience of hermaphroditic species to
**Table 1  Definitions of life history and spawning behavior parameters included in the analysis.**

| Parameter Type | Parameter | Abbreviation | Weighted influence on vulnerability | Description |
|---|---|---|---|---|
| **Spawning Behavior** | Aggregating Behavior (0–4) | Agg | 0.261 | The degree of aggregating behavior associated with spawning: does not aggregate = 0; simple migratory spawner = 1; resident spawning aggregation = 2; mixed resident and transient aggregations = 3; transient spawning aggregation = 4. |
| | Spawning Season Duration (1–12) | Duration | 0.215 | Number of months that the species spawns, with shorter spawning seasons conferring higher vulnerability to aggregation fishing. |
| | Relative Abundance (1–6) | Rel. Ab. | 0.232 | Change in abundance of fish relative to non-reproductive periods. No change in abundance between spawning and non-spawning periods = 1; abundance doubles from solitary to few to ca. 10 fish (clustering of polygynous groups) = 2; abundance increases from small groups to 100–200 fish = 3; abundance increases from small groups to 500–1,000 fish = 4; abundance increases from small groups to 1,000–10,000 fish =5; abundance increases from small groups to >10,000 fish = 6. Larger abundance changes confer higher vulnerability to aggregation fishing. |
| **Life History** | Max Age (years) | $A_{max}$ | 0.293 | Maximum age in years |
| | Max Weight (kg) | $W_{max}$ | | Maximum weight in kilograms |
| | Max Length (cm) | $L_{max}$ | | Maximum reported length for the species in centimeters |
| | k (vB Growth Coefficient) | $k$ | | von Bertalanffy growth coefficient |
| | $L_{inf}$ (Asymptotic Length, cm) | $L_{inf}$ | | Asymptotic length for von Bertalanffy growth equation, expressed in centimeters |
| | Age at Maturity (months) | $A_m$ | | Age at 50% maturity in months |
| | Length at Maturity (cm) | $L_m$ | | Length at 50% maturity in centimeters |
| | M (Natural Mortality) | $M$ | | Death rate per year not associated with fishing |

fishing, often in complex ways (*Robinson et al., 2017*; *Schram & Steele, 2020*), and should thus be the focus of separate study.

Spawning behavior was characterized by the degree of aggregating behavior, spawning season duration in months, and the estimated magnitude of change in relative abundance of fish during peak spawning periods relative to non-reproductive periods (Table 1). The duration of the spawning season was determined by the number of months that spawning was reported to occur in the GOM based primarily on the sampling of mature females and assessments of their reproductive condition (e.g., elevated gonadosomatic index levels or the presence of actively spawning females) (*Brown-Peterson et al., 2011*; *Lowerre-Barbieri et al., 2011*). However, some fish aggregate at spawning sites over a longer time period than active spawning occurs (*Heyman et al., 2005*; *Heyman et al., 2019*).

The phrase "during spawning" refers to the spawning season and was used in this context throughout the paper.

In all cases, the values for each category were specific to the GOM, as they may vary among populations and regions along a species' geographic range (*Lowerre-Barbieri et al., 2009*; *Heyman et al., 2019*). In instances where the spawning season varied within the GOM, we incorporated the entire range of spawning months. If data related to the degree to which a species formed spawning aggregations was not available for the GOM (e.g., *Epinephelus flavolimbatus*), we determined their behavior based on literature from the Southeast U.S. and Caribbean (see Results and references in Material S3). Aggregation type was intended to reflect the distance traveled to a spawning site, and the number/distribution of spawning sites, reflected as the degree to which the species aggregates to spawn on a scale of 0–4. Species that do not aggregate to spawn were scored 0, simple migratory spawners were scored 1, species that form resident aggregations were scored 2, species that form resident and transient aggregations (*i.e.*, mixed) were scored 3, and transient aggregations were scored 4. The estimated change in relative abundance was based on order of magnitude comparisons (e.g., 1×, 10×, 100×) between peak spawning times and abundance during non-spawning periods. The scale (1–6) distinguished among species that are solitary, grouping, or schooling for non-reproductive functions. The spawning behavior categories were ordinal in this case, because they related to varying degrees of vulnerability to fishing pressure (*Robinson & Samoilys, 2013*; *Robinson, 2015*). Spawning season duration and the degree of aggregating behavior was obtained exclusively through a comprehensive literature review. Relative abundance was based on expert judgement of the authors and collaborators associated with this project.

## Correlation analysis and PCA

Spearman's Rank Correlation was used to explore relationships between life history traits and spawning behavior parameters in R (*R Core Team, 2016*), tested at a significance level of $\alpha = 0.05$. The parameters were also used to perform Principal Component Analysis (PCA) using a correlation matrix with normalized data. We could not identify reliable estimates of $M$ for the range of species of interest and thus did not integrate this trait into the PCA. A PCA was chosen rather than a Linear Discriminant Analysis (LDA) or cluster analysis, because we were interested in the continuum of reproductive behaviors and life history traits. Although PCA and k-means clustering are closely related, the PCA offers a continuous solution rather than clusters of homogenous groups (*Ding & He, 2004*). Likewise, PCA is unsupervised and finds the directionality of maximum variance, where LDA maximizes class separability. The results of the PCA were represented with a biplot along with the stock status based on region-wide assessments in the GOM and the designations of NOAA Fisheries (https://www.fisheries.noaa.gov). "Not overfished" species were those that have never been designated as overfished. "Overfished" species included those that are currently designated or had previously been designated at some point. For example, Red Grouper were placed in the category of overfished, because previous assessments had made that designation even while newer criteria concluded that the stock has never been overfished during the time series (*SEDAR, 2019*). "Unassessed"
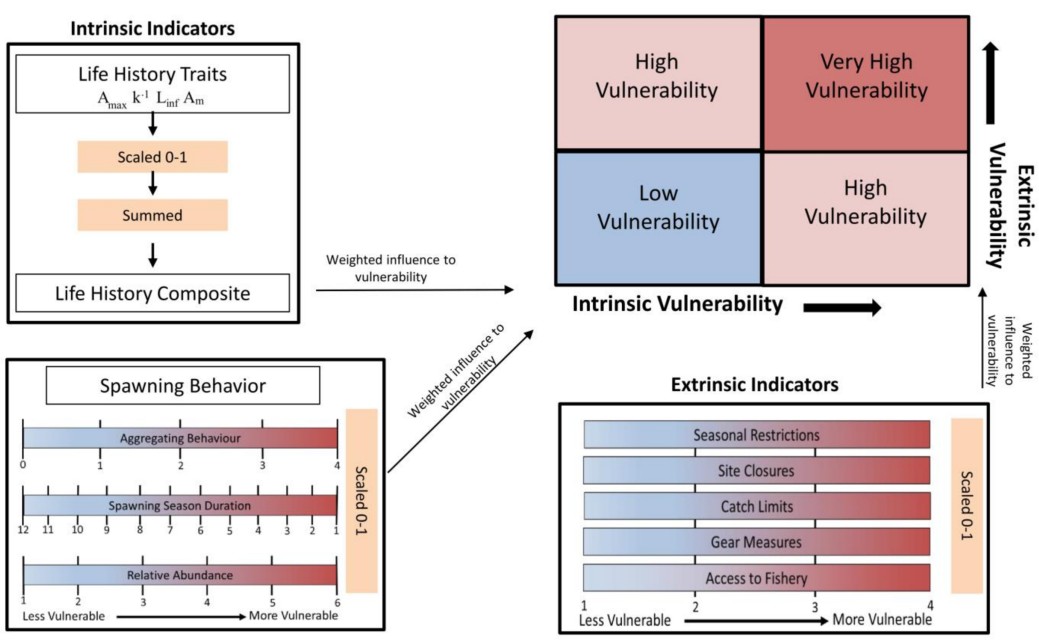

**Figure 1** Flow chart illustrating the process of calculating intrinsic and extrinsic vulnerability scores. Definitions for each of the indicators, the scales used, and the weighted vulnerabilities are in Table 1 and Table 2.

species were those that have not been assessed in the GOM. Harvest of Nassau Grouper (*Epinephelus striatus*), Atlantic Goliath Grouper (*Epinephelus itajara*), and Red Drum (*Sciaenops ocellatus*) is prohibited, because each of these stocks were historically overfished, and thus each were classified as overfished. Gag (*Mycteroperca microlepis*) and Red Grouper (*Epinephelus morio*) have also been previously designated as overfished and were labelled as such, although their current stock status is not overfished. There are no region-wide assessments for the coastal species, only limited state assessments, so those stocks were considered unassessed. A one-tailed t-test was used to compare the average PC1 and PC2 score between overfished and not overfished stocks to identify common traits among the different stock statuses. A one-tailed test was used, because each PC has directionality that is related to the theoretical vulnerability to overfishing. Lower PC1 scores and higher PC2 scores would indicate higher vulnerability.

## Vulnerability analysis

Figure 1 contains a flow chart illustrating the process of the vulnerability assessment. The vulnerability analysis was based on previous studies that accounted for life history traits and spawning behaviors associated with vulnerability to fishing during spawning (*Cheung, Pitcher & Pauly, 2005*; *Robinson & Samoilys, 2013*; *Robinson, 2015*). The indicators used are split between two axes and include intrinsic indicators (life history and behavior of the species) and extrinsic indicators (behavior of the fishery and management) that measure the susceptibility and exposure of spawning fish to fishing. The intrinsic indicators included the degree of aggregating behavior associated with spawning, duration of the spawning season, change in relative abundance during spawning, $A_{max}$, $L_{inf}$, $A_m$,

**Table 2 Definitions and categories of the extrinsic vulnerability indicators used in the analysis.**

| Parameter | Weighted influence on vulnerability | Description |
| --- | --- | --- |
| Access to Fishery (1–4) | 0.081 | Extent to which access to the fishery is limited. 4 = open (basic commercial/recreational license), 3 = reef fish permit (comm. regulation) and or charter/headboat reef fish and coastal pelagics permit (rec. regulation), 2 = reef fish permit & IFQ program, 1 = closed |
| Catch Limits (1–4) | 0.215 | Number of catch limits, 4 = 0–1 regulation, 3 = 2–3 regulations, 2 = 4 regulations, 1 = 5 regulations |
| Gear Measures (1–4) | 0.114 | Amount of restrictions on gear used, 4 = 9 or more allowable gear types, 3 = 6–8 allowable gear types, 2 = 3–5 allowable gear types, 1 = 0–2 allowable gear types |
| Seasonal Restrictions (1–4) | 0.291 | Spawning season prohibition of take or trade. 4 = none, 3 = seasonal closure, not during spawning, 2 = seasonal closure during peak spawning, 1 = closed during entire spawning season |
| Site Closures (1–4) | 0.300 | Spatial closure of spawning site. 4 = no regulations, 3 = restricted gear, 2 = site closed part of the year, 1 = site closed all year |

and $k$, as their correlation to fishing vulnerability has been previously illustrated. The extrinsic indicators reflect the degree to which spawning fish are protected within state or federal waters of the GOM and included access to the fishery, catch limits, gear measures, seasonal closures and site closures during the spawning season (Table 2).

The extrinsic indicators were each scored on a scale of 1–4 with a larger number denoting a higher vulnerability. Federal regulations were considered for all species except for coastal species that are primarily targeted in state waters. For those species, state regulations were considered since they inhabit, spawn, and are fished in state waters. The assigned scores for coastal species reflected the average score among the Gulf states (Texas, Louisiana, Mississippi, Alabama, and Florida). Access to the fishery captured the extent to which access is restricted via a regulated number of permits and the individual fishing quota (IFQ) program (1), or open, requiring a basic commercial or recreational license (4). Catch limits included minimum and maximum size limits as well as daily bag limits and quotas for the commercial and recreational fishery. Scores ranged from 1 (for a total of 5 catch limits) to 4 (for no catch limits). Gear measures indicated the restrictions on gear types used in the fishery and ranged from 0 to 2 allowable gear types (1), to 9 or more allowable gear types (4). Seasonal restrictions reflected the level of spawning season prohibition of take, from prohibition during the entire spawning season (1) scaling to no restrictions (4). The selected value was the least restrictive score of recreational and commercial seasonal restrictions. Site closures ranged from total spatial closure of spawning site (1) scaling to no spatial closures (4). Scores indicated if spawning sites were closed all year or involved a complete fishery closure (1), more than or equal to 10% of known spawning sites were protected by complete or seasonal site closures (2), less than 10% of known spawning sites were protected by complete or seasonal site closures (3), or if there were no spawning site closures or the species did not reproduce in federal waters (*i.e.*, coastal species) (4). State site closures are few and spatially minimal. Therefore, all coastal species received a score of 4 (*i.e.*, no site-based regulations).

Reciprocal values were used for spawning season duration and k to preserve the direction of their influence on, and association with, increasing vulnerability. The values for life history traits ($A_{max}$, $L_{inf}$, k, and $A_m$) were combined into one category as a life history composite. The values for each intrinsic and extrinsic indicator were scaled 0–1. Each parameter was then weighted according to the impact and relative influence on vulnerability as determined in *Robinson (2015)*, which assigned the weights through an Analytic Hierarchy Process (*Saaty, 1987*). The process involved ranking each parameter in terms of its influence on vulnerability and performing pair-wise comparisons to develop a matrix of indicator weights with the final value calculated as the average among the matrices. Then, the predictive ability of those weighted parameters were validated against the status of global fisheries targeting spawning aggregations (*Robinson & Samoilys, 2013*). The validation of the vulnerability index showed the correlation of the parameters with abundance trends in seven species of reef fishes, and ultimately supported the use of the indicator-based framework (*Robinson & Samoilys, 2013*). The respective weights for the intrinsic and extrinsic indicators are in Tables 1 & 2. The values were summed for each group of indicators to obtain an intrinsic and extrinsic vulnerability score. The scores along the two axes were combined to get an overall index of vulnerability to fishing during spawning, which was calculated as the Euclidian distance from the origin (position in bivariate space). Equal weight was given to intrinsic and extrinsic indicators as those compound factors were rescaled (0–1) to provide a relative comparison among the selected species.

# RESULTS

## Species selection

The species selection process identified 24 species to be included in the final assessment. Four common coastal and estuarine species were added to the analysis post hoc, because they are important to fisheries in state waters throughout the GOM: Spotted Seatrout (*Cynoscion nebulosus*), Sheepshead (*Archosargus probatocephalus*), Southern Flounder (*Paralichthys lethostigma*), and Black Drum (*Pogonias cromis*). Although the selected species are not a random sample, they represent a wide range of exploited species from which to examine life history traits and spawning behavior in relation to overfishing and vulnerability, which was our objective. Although aggregating behavior in association with spawning was a main focus of the analysis, six species that do not aggregate to spawn were included among the selected species. Highly migratory, schooling, pelagic species (e.g., tunas) were excluded, because of the vastly different reproductive habitats of open ocean pelagic species. As a group, they are well studied and managed as a separate unit. This resulted in a final list of 28 species to be analyzed, which are listed on Table 3 and described in greater detail in the Material S1.

## Life history information and spawning behavior

A total of 801 documents including peer-reviewed literature, grey literature, and SEDAR reports were reviewed for spawning and life history information on the 28 species. The values for each parameter are presented in Table 3, and a full table with citations for

**Table 3 Spawning behavior and life history characteristics for the 28 species analyzed.**

| Common Name | Agg. (0-4) | Duration (# mos) | Rel. Ab. (1-6) | $A_{max}$ (yr) | $W_{max}$ (kg) | $L_{max}$ (cm) | $k$ | $L_{inf}$ (cm) | $A_m$ (mos) | $L_M$ (cm) | $M$ |
|---|---|---|---|---|---|---|---|---|---|---|---|
| Gray Triggerfish | 2 | 4 | 4 | 15 ± 1 | 6 | 30 | 0.14 ± 0.06 | 59 ± 3 | 18 ± 2 | 17 ± 3 | 0.27 |
| Almaco Jack | 3 | 8 | 3 | 22 | 60 | 160 | 0.13 | 163 | 53 | 81 | N/A |
| Greater Amberjack | 3 | 4 | 3 | 15 ± 1 | 81 | 190 | 0.14 ± 0.01 | 144 ± 7 | 27 ± 2 | 79 ± 4 | 0.25 ± 0.03 |
| Black Grouper | 4 | 5 | 4 | 33 ± 0 | 163 | 150 | 0.14 ± 0.01 | 133 ± 1 | 78 ± 4 | 86 ± 2 | 0.14 ± 0.02 |
| Gag | 4 | 4 | 3 | 31 ± 3 | 37 | 145 | 0.13 ± 0.01 | 128 ± 1 | 42 ± 2 | 54 ± 3 | 0.13 ± 0.01 |
| Atlantic Goliath Grouper | 4 | 5 | 3 | 37 ± 4 | 363 ± 68 | 250 | 0.09 ± 0.01 | 222 ± 6 | 72 ± 6 | 120 ± 3 | 0.12 ± 0.03 |
| Nassau Grouper | 4 | 3 | 6 | 29 ± 2 | 27 ± 0 | 100 | 0.13 ± 0.02 | 76 ± 3 | 60 ± 0 | 40 ± 2 | 0.18 ± 0.03 |
| Red Grouper | 0 | 5 | 2 | 29 ± 1 | 23 | 125 | 0.13 ± 0.02 | 83 ± 2 | 34 ± 6 | 29 ± 4 | 0.14 ± 0.02 |
| Scamp | 3 | 6 | 3 | 31 ± 1 | 13 | 107 | 0.09 ± 0.00 | 77 ± 9 | 24 ± 3 | 33 ± 1 | 0.15 ± 0.09 |
| Warsaw Grouper | 3 | 8 | 3 | 41 ± 0 | 198 ± 4 | 235 | 0.05 ± 0.00 | 239 ± 0 | 49 | 81 | N/A |
| Yellowedge Grouper | 3 | 10 | 3 | 85 ± 10 | 20 ± 1 | 115 | 0.06 ± 0.01 | 100 ± 3 | 96 ± 0 | 55 ± 9 | 0.07 ± 0.11 |
| Yellowfin Grouper | 4 | 8 | 4 | 15 ± 1 | 19 ± 0 | 100 | 0.12 ± 0.02 | 89 ± 6 | 44 | 54 ± 10 | 0.26 ± 0.05 |
| Yellowmouth Grouper | 4 | 12 | 3 | 28 ± 3 | 9 | 84 | 0.08 ± 0.01 | 83 ± 1 | 36 ± 11 | 43 ± 1 | 0.23 ± 0.02 |
| Speckled Hind | 0 | 3 | 1 | 45 ± 3 | 30 | 110 | 0.12 ± 0.01 | 89 ± 5 | 79 ± 9 | 53 ± 8 | 0.15 ± 0.01 |
| Snowy Grouper | 0 | 10 | 1 | 35 ± 2 | 30 | 122 | 0.09 ± 0.01 | 106 ± 5 | 60 ± 2 | 60 ± 4 | 0.19 ± 0.02 |
| Hogfish | 2 | 8 | 2 | 23 ± 1 | 10 ± 2 | 91 | 0.11 ± 0.08 | 85 ± 6 | 11 ± 2 | 15 ± 1 | 0.18 ± 0.11 |
| Cubera Snapper | 4 | 4 | 6 | 22 ± 1 | 57 ± 8 | 160 | 0.16 ± 0.01 | 120 ± 7 | 24 ± 9 | 62 ± 6 | 0.15 ± 0.08 |
| Mutton Snapper | 4 | 4 | 5 | 40 ± 6 | 16 ± 3 | 94 | 0.17 ± 0.01 | 86 ± 4 | 48 ± 3 | 50 ± 5 | 0.11 ± 0.03 |
| Red Snapper | 2 | 5 | 2 | 48 ± 4 | 23 ± 2 | 100 | 0.19 ± 0.05 | 86 ± 4 | 24 ± 3 | 23 ± 2 | 0.1 ± 0.02 |
| Vermilion Snapper | 2 | 6 | 2 | 26 ± 2 | 3 | 63 | 0.33 ± 0.04 | 34 ± 2 | 24 ± 4 | 14 ± 3 | 0.25 ± 0.03 |
| Tilefish | 0 | 6 | 2 | 40 ± 3 | 26 | 125 | 0.13 ± 0.01 | 83 ± 3 | 24 ± 2 | 34 ± 5 | 0.13 ± 0.01 |
| Southern Flounder | 4 | 4 | 5 | 8 ± 1 | 9 | 92 | 0.28 ± 0.08 | 65 ± 6 | 24 ± 5 | 40 | 0.36 |
| Black Drum | 3 | 7 | 4 | 58 ± 8 | 51 | 150 | 0.17 ± 0.02 | 114 ± 21 | 60 | 65 ± 2 | 0.06 ± 0.00 |
| Red Drum | 3 | 4 | 4 | 42 ± 2 | 45 | 160 | 0.32 ± 0.04 | 88 ± 7 | 48 ± 8 | 68 ± 6 | 0.16 ± 0.04 |
| Spotted Seatrout | 2 | 6 | 3 | 12 ± 1 | 8 | 70 | 0.32 ± 0.05 | 69 ± 3 | 12 ± 0 | 23 ± 1 | 0.3 |
| Gray Triggerfish | 1 | 6 | 2 | 24 ± 2 | 42 | 184 | 0.19 ± 0.02 | 115 ± 5 | 48 | 60 ± 12 | 0.17 ± 0.01 |
| Almaco Jack | 1 | 6 | 2 | 11 ± 2 | 6 | 101 | 0.61 ± 0.03 | 56 ± 3 | 8 ± 0.01 | 30 ± 2 | 0.3 |
| Greater Amberjack | 4 | 3 | 5 | 20 ± 2 | 10 | 92 | 0.36 ± 0.03 | 46 ± 1 | 24 ± 6 | 30 | 0.15 |

**Note:**

Values are specific to the GOM and reflect the average (±SE). Definitions of each characteristic are in Table 1. An annotated table with citations for each entry is available online (http://geo.gcoos.org/restore/).

each entry is available online (http://geo.gcoos.org/restore/) and in Material S3. Values for *M* were not found for Almaco Jack (*Seriola rivoliana*) or Warsaw Grouper (*Hyporthodus nigritus*).

The degree of aggregating behavior was determined directly from descriptions in the literature for the GOM for all species except for Warsaw Grouper and Yellowedge Grouper (*Hyporthodus flavolimbatus*), which were classified by the authors' expert opinions based on studies from the Southeast U.S. and Caribbean (Table 3, Material S3). Based on the literature review, 10 of 28 species were determined to form transient aggregations. Groupers (Epinephelidae; *n* = 6;) were the most common family listed in the transient group. Seven species were categorized as forming mixed aggregations and included three

grouper species, the two jacks (Carangidae) and two sciaenid (Sciaenidae) species. Five species were determined to form resident aggregations. The two mackerel (*Scomberomorus*) species were designated as simple migratory spawners. Red Grouper, Snowy Grouper (*Hyporthodus niveatus*), Speckled Hind (*Epinephelus dummondhayi*), and Tilefish were determined to not form spawning aggregations based on the lack of any evidence of such behavior in the literature for the GOM or elsewhere.

The change in relative abundance of fish during spawning was greatest for Cubera Snapper (*Lutjanus cyanopterus*) and Nassau Grouper, which corresponded to aggregations of >10,000 fish (Table 3). Three additional species were classified as 5, corresponding to aggregations of 1,000–10,000 fish. Most species (*n* = 9) were scored 3, indicating that aggregations were composed of small groups of 100–200 fish. Seven species were scored a 2, corresponding to small groups or doubling of abundance relative to non-reproductive periods.

Spawning seasons ranged from 3 to 12 months, with grouper species having the largest variation in spawning season (Table 4). Yellowedge Grouper had the most protracted spawning seasons at 12 months followed by Yellowmouth Grouper (*Mycteroperca interstitialis*) at 10 months, while Nassau Grouper and Sheepshead only spawned three months out of the year. The greatest number of species spawned in June (*n* = 21), with an average of 18 species spawning per month, April through August. The fewest number of species spawned in November (*n* = 7) and December (*n* = 4). The snappers were the only family that showed consistency in spawning season, with peak spawning occurring June through August. An annotated table with all references is available online (http://geo.gcoos.org/restore/) and in Material S3.

## Correlation analysis and PCA

Aggregation type was positively correlated with relative abundance ($r_s$ = 0.819, $p$ < 0.01) (Fig. 2, Table S2.1). Spawning season duration was negatively correlated with relative abundance ($r_s$ = −0.538, $p$ = 0.01) and $k$ ($r_s$ = −0.526, $p$ = 0.01). We found no other significant relationships between spawning behavior parameters and life history traits. However, most of the life history traits were significantly correlated with each other (Fig. 2, Table S2.1). As expected, the maximum growth parameters (age, weight, length, $L_{inf}$) were positively correlated with each other and $A_m$, and negatively correlated with $k$ and $M$.

The first two axes of the PCA explained 64.4% of the variation in the data (Fig. 3, Table S2.2). Along PC2, the eigenvectors for spawning behavior characteristics had a greater influence (as defined by the absolute value of the eigenvector) on the distribution of species than life history traits. Relative abundance (0.634) was the greatest followed by aggregation type (0.493) and spawning season duration (−0.481). Along PC1, the eigenvectors for life history traits had a greater influence than those for spawning behaviors. $L_m$ was the greatest (−0.440), followed by $L_{inf}$ (−0.430), $L_{max}$ (−0.405), and $W_{max}$ (−0.399). The PCA biplot (Fig. 3) illustrates the separation of species by the reproductive and life history parameters along the first two PC axes within the context of stock status. Six of the species have been deemed not overfished. Ten species have been overfished, including Atlantic Goliath Grouper and Nassau Grouper, which are both

**Table 4 Spawning season of 28 species in the Gulf of Mexico, sorted by family.**

| Family | Common Name | Jan | Feb | Mar | Apr | May | Jun | Jul | Aug | Sep | Oct | Nov | Dec |
|---|---|---|---|---|---|---|---|---|---|---|---|---|---|
| Epinephelidae | Black Grouper | ■ | ■ | ■ | ▓ | | | | | | | | ▓ |
| Epinephelidae | Gag | ▓ | ■ | ■ | ▓ | | | | | | | | |
| Epinephelidae | Atlantic Goliath Grouper | | | | | | ▓ | ■ | ■ | ■ | ▓ | | |
| Epinephelidae | Nassau Grouper | ■ | ■ | | | | | | | | | | ■ |
| Epinephelidae | Red Grouper | | ▓ | ■ | ■ | ■ | ▓ | | | | | | |
| Epinephelidae | Scamp | ▓ | ▓ | ■ | ■ | ▓ | ▓ | | | | | | |
| Epinephelidae | Snowy Grouper | | | | ▓ | ▓ | ▓ | ▓ | ▓ | ▓ | ▓ | | |
| Epinephelidae | Speckled Hind | | | | ▓ | ▓ | ▓ | ▓ | ▓ | ▓ | | | |
| Epinephelidae | Warsaw Grouper | | | | ▓ | ▓ | ▓ | ▓ | ▓ | ▓ | ▓ | ▓ | |
| Epinephelidae | Yellowedge Grouper | | ▓ | ■ | ■ | ■ | ■ | ■ | ■ | ■ | ▓ | ▓ | |
| Epinephelidae | Yellowfin Grouper | ▓ | ▓ | ■ | ■ | ■ | ▓ | ▓ | ▓ | | | | |
| Epinephelidae | Yellowmouth Grouper | ▓ | ▓ | ▓ | ■ | ■ | ▓ | ▓ | ▓ | ▓ | ▓ | ▓ | ▓ |
| Lutjanidae | Cubera Snapper | | | | | | ▓ | ■ | ■ | ▓ | | | |
| Lutjanidae | Mutton Snapper | | | | | ■ | ■ | ▓ | ▓ | | | | |
| Lutjanidae | Red Snapper | | | | | ▓ | ■ | ■ | ■ | ▓ | | | |
| Lutjanidae | Vermilion Snapper | | | | ▓ | ▓ | ■ | ■ | ■ | ▓ | | | |
| Sciaenidae | Black Drum | ▓ | ■ | ■ | ▓ | | | | | ▓ | ▓ | ■ | |
| Sciaenidae | Red Drum | | | | | | | | | ▓ | ■ | ■ | ▓ |
| Sciaenidae | Spotted Seatrout | | | | ▓ | ■ | ■ | ■ | ■ | ▓ | | | |
| Scombridae | King Mackerel | | | | | ▓ | ■ | ■ | ■ | ■ | | | |
| Scombridae | Spanish Mackerel | | | | ▓ | ■ | ■ | ▓ | ▓ | ▓ | | | |
| Carangidae | Almaco Jack | | | | ▓ | ▓ | ▓ | ▓ | ▓ | ▓ | ▓ | | |
| Carangidae | Greater Amberjack | | | ■ | ■ | ■ | ▓ | | | | | | |
| Sparidae | Sheepshead | | ▓ | ■ | ■ | | | | | | | | |
| Paralichthyidae | Southern Flounder | ▓ | | | | | | | | | ▓ | ■ | ■ |
| Malacanthidae | Tilefish | ▓ | ▓ | ▓ | ■ | ▓ | ▓ | | | | | | |
| Labridae | Hogfish | ■ | ■ | ■ | ■ | ▓ | ▓ | | | | | ▓ | ■ |
| Balistidae | Gray Triggerfish | | | | | ▓ | ■ | ■ | ▓ | | | | |

**Note:**
Grey indicates the extent of the spawning season; black indicates the peak spawning months.

closed to recreational and commercial fishing, as well as Red Drum, which is closed to fishing in federal waters and to commercial fishing in most states along the GOM. Twelve species have not been assessed, including three coastal species. The PC2 scores were

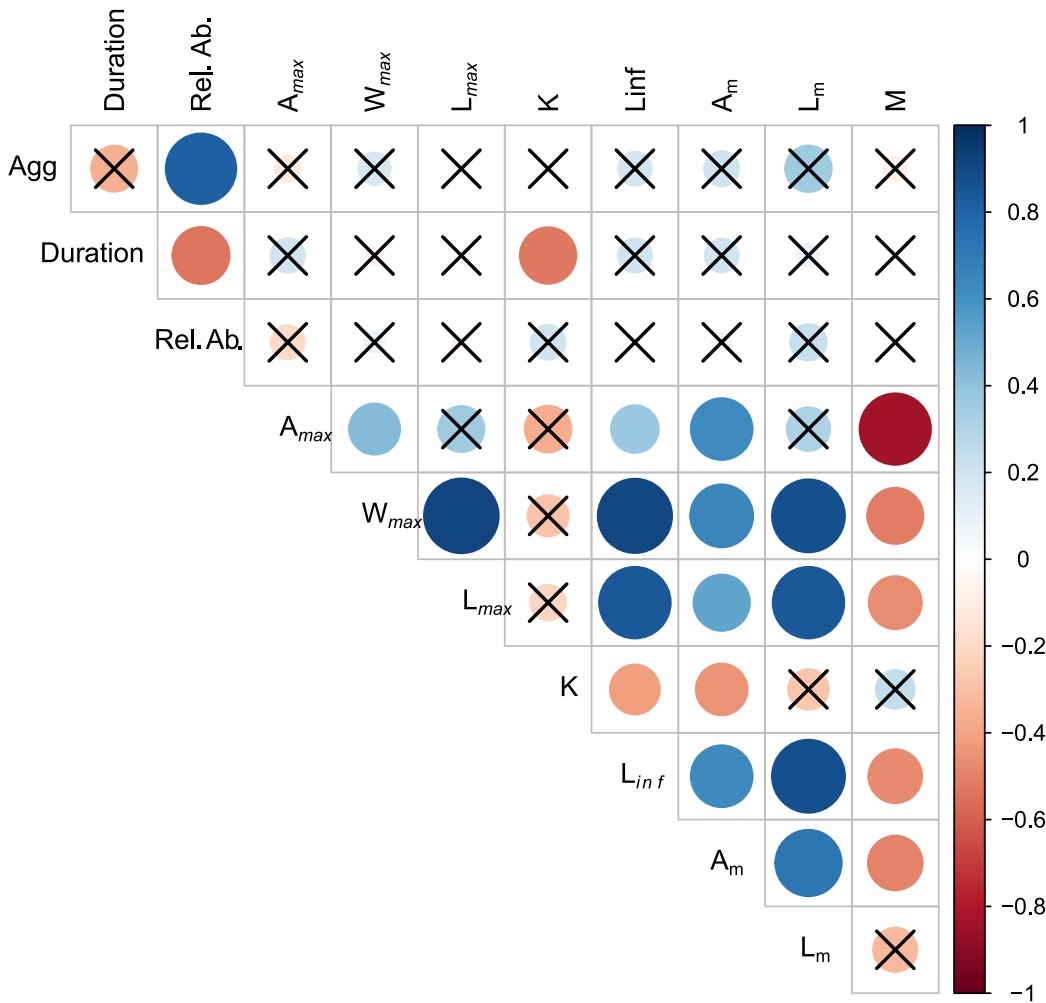

**Figure 2  Visualization of correlation matrix of spawning behavior and life history parameters.** Blue indicates positive correlation and red reflects negative correlation. The size of the circle reflects the value of Spearman's rank correlation coefficient and boxes without an "x" are significant ($\alpha$ = 0.05).

positive for 7 out of 10 species that are currently or historically overfished whereas they were negative for 4 of the 6 not overfished species. The mean PC score was significantly higher for stocks currently or historically overfished than not overfished for PC2 (t = 2.03, df = 9.25, $p$ = 0.04) but not for PC1 (t = −0.844, df = 15, $p$ = 0.79) (Fig. 4). Overfished status was closely associated with positive changes in relative abundance and aggregation type, and negatively related to spawning season duration. Peak spawning months was included in an additional PCA, but it did not change the output and had very low eigenvectors, so it was not considered further (Fig. S2.1).

## Vulnerability analysis
Scores for the intrinsic and extrinsic vulnerability analysis along with the extrinsic indicator scores for federal and state management regulations are available in Material S4. Sheepshead and Southern Flounder had the two greatest overall vulnerability scores,

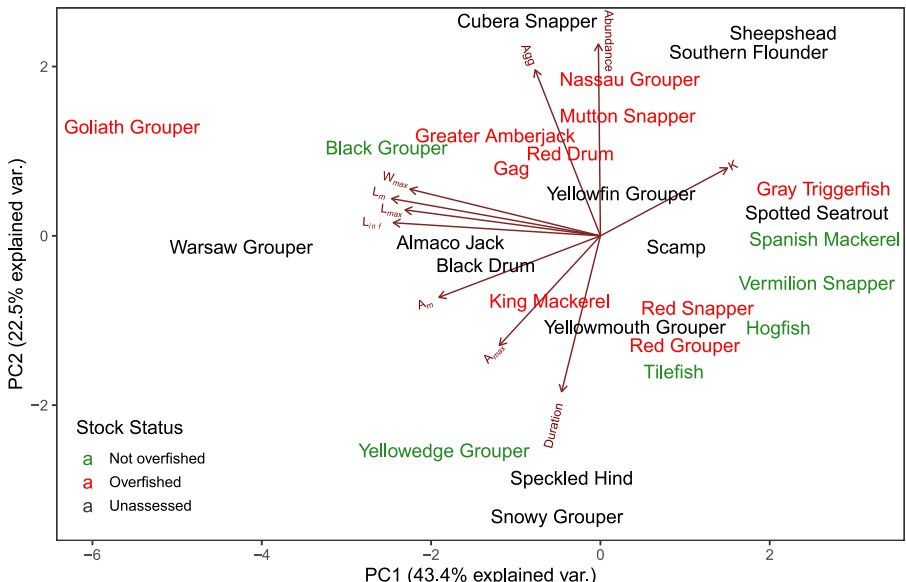

**Figure 3 Biplot of PCA showing the first two principal components.** The arrows show the relative loadings of each PC axis and the color of the species indicates the stock status. Historically overfished species (Atlantic Goliath Grouper, Nassau Grouper, Red Drum, Gag Grouper, and Red Grouper) have been labelled overfished in addition to those species that are currently designated as overfished.

but neither are federally managed nor have they been assessed at the region-wide level (Fig. 5). Cubera Snapper had the greatest vulnerability scores among the federally managed species, but the stock has not been assessed. All species scored high in extrinsic vulnerability (>0.5), except for Gray Triggerfish (*Balistes capriscus*) and the three species with closed fisheries (Atlantic Goliath Grouper, Nassau Grouper, Red Drum) that also had the lowest overall vulnerability scores. The intrinsic vulnerability score was greatest for Nassau Grouper and least for Spanish Mackerel (*Scomberomorus maculatus*). Cubera Snapper, Warsaw Grouper, and Yellowmouth Grouper had the highest intrinsic vulnerability scores among the unassessed species.

## DISCUSSION

Spawning behavior represents a separate and distinct aspect of fish ecology that is important to consider for accurate predictions of vulnerability and resilience in exploited stocks (*Erisman et al., 2017a*, *2017b*; *Lowerre-Barbieri et al., 2017*). Our results show that characteristics of spawning behavior known to be associated with vulnerability to fishing are not directly related to life history traits that are typically associated with vulnerability in 28 species of exploited marine fishes in the GOM. Using PCA analysis, we demonstrated that the characterization of spawning behaviors improved the identification of vulnerable and overfished species more than "traditional" life history traits alone. We found that nearly all species showed susceptibility to overfishing or becoming overfished during their spawning season due to very few state or federal regulations to

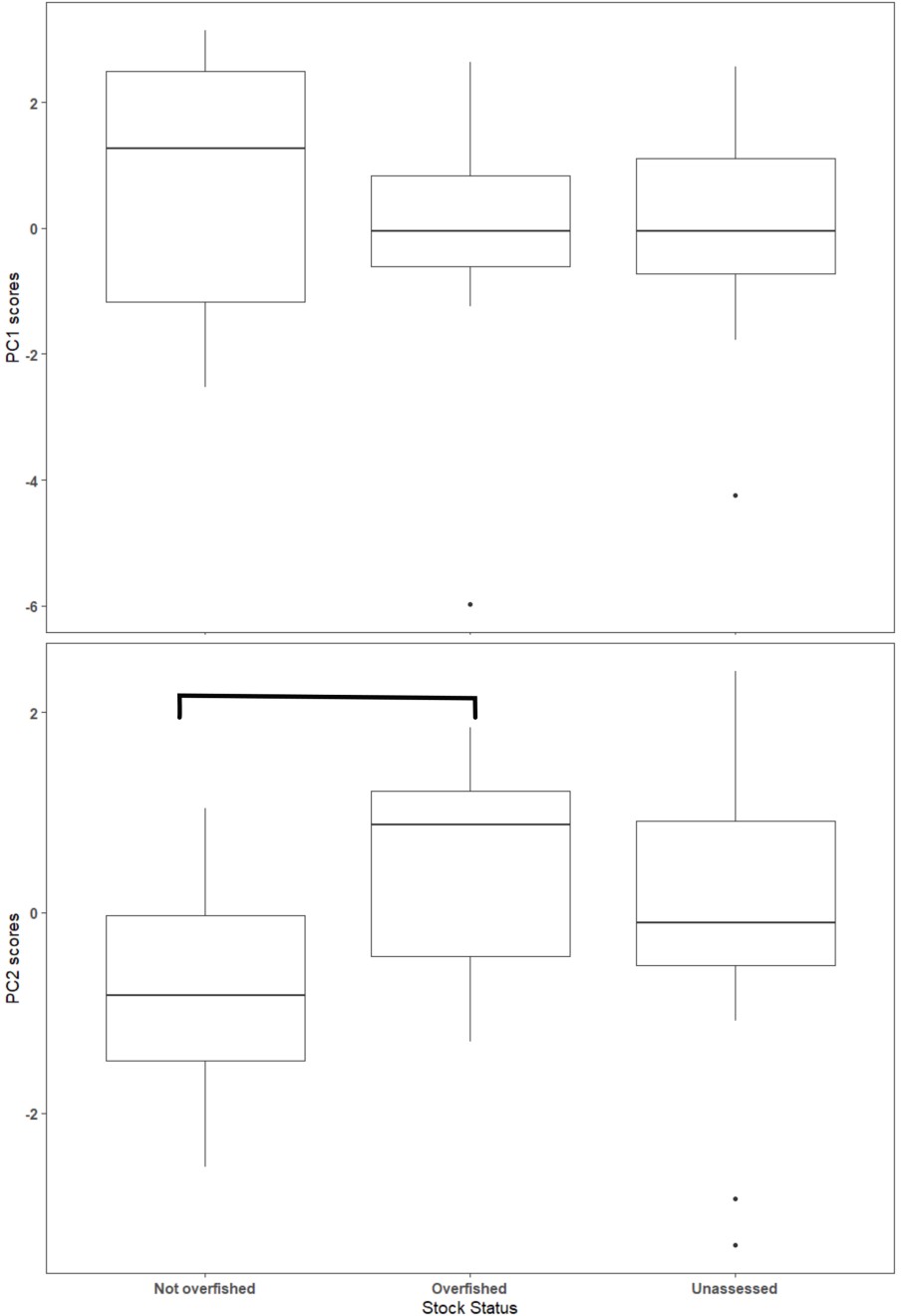

**Figure 4 Box plot of Principal Component (PC) scores by stock status.** The horizontal line within the box represents the mean value and the box outlines the interquartile range (IQR). Whiskers indicate the highest and lowest value within 1.5 * IQR, values outside that range are outliers and are plotted as points. Overfished and not overfished on PC2 are significantly different (t = 1.88, df = 13.4, $p$ = 0.08). Statistical comparisons were only made between overfished and not overfished scores.

protect spawning fish in the GOM. However, there was a large range of intrinsic vulnerability based on the diversity of spawning behavior and life history traits exhibited among the species studied. Increased effort to understand patterns of spawning behavior

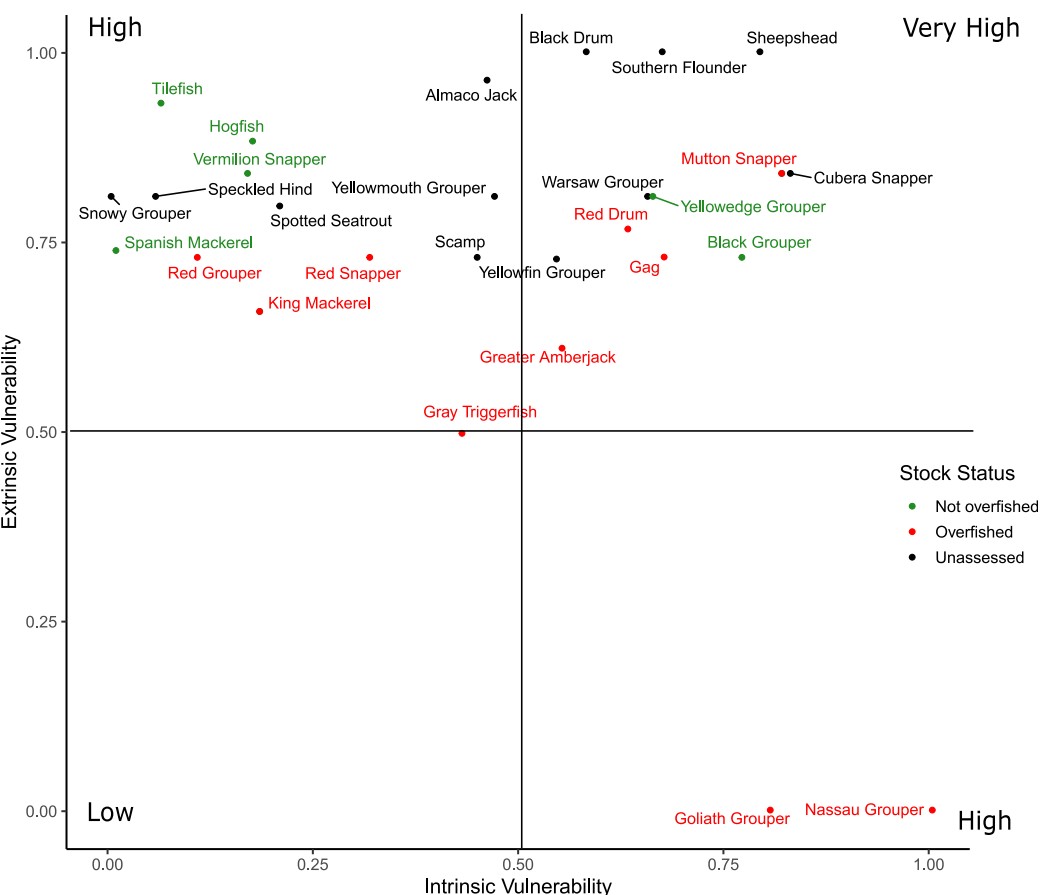

**Figure 5** **Distribution of the overall vulnerability index of the 28 species.** The position of each species within each quadrant indicates their relative vulnerability from low to very high.

and the distribution of fishing effort and catch in relation to spawning would aid conservation and management efforts in the GOM. Specifically, it would help identify species that are particularly vulnerable to fishing during spawning and support the enactment of protection measures to enhance resilience to fisheries exploitation (*Erisman et al., 2018*; *Heyman et al., 2019*). Consideration of spawning dynamics in addition to more traditional life history traits, management, and fisher behavior would help focus monitoring, research, and rebuilding efforts on the most vulnerable species. In the U.S. GOM, stock assessments generate a probability distribution function of overfishing limits that is converted to allowable biological catch based on the Gulf Council's risk tolerance for scientific uncertainty. Our findings imply a larger buffer may be necessary to avoid overfishing for transient aggregating species, especially when this spawning behavior is not explicitly considered in stock assessment.

There was a clear separation between life history traits and spawning behavior with respect to loadings along the first two principal component axes. The component scores along PC1 separated species based on life history traits, and PC2 was primarily loaded with spawning behaviors. Spawning behavior provided a better distinction between

overfished and not-overfished species than the life history traits (*i.e.*, more status information can be derived from PC2 than PC1). Stock status of a fishery is influenced by multiple factors, which include extrinsic components such as fishing practices (e.g., fishing effort; gear efficiency and selectivity), management actions, stock assessment uncertainty stemming from data limitations in the indicators (e.g., biomass), and changes in biological reference points (*Rosenberg & Restrepo, 1994*; *Branch et al., 2011*). We acknowledge that stock status may not completely reflect the vulnerability and resilience of a species, but it is commonly assumed to be a function of life history traits and historical exploitation (*Costello et al., 2012*). In addition, stock status provides a metric to compare the relationships between exploitation, life history and reproductive behavior, especially with respect to identifying species that might be at risk of overfishing. Several unassessed stocks with high intrinsic and extrinsic vulnerability were identified, including deep-water grouper stocks such as Speckled Hind and Warsaw Grouper. In the nearby U.S. South Atlantic region, spatial protections for these stocks have been considered, to reduce post-release mortality (*Farmer & Karnauskas, 2013*) and then implemented by creating a declaration of five new Spawning Special Management Zones (sSMZs) in which bottom fishing is restricted (*South Atlantic Fisheries Management Council, 2017*).

As has been shown in several review studies on spawning aggregations (e.g., *Erisman et al., 2011*; *Sadovy De Mitcheson & Erisman, 2012*), species with spawning behaviors characterized by transient aggregations (that form over short durations and have large changes in relative abundance) were more likely to be overfished than those that form resident aggregations, mixed aggregations, or do not aggregate for spawning. Therefore, the consideration of spawning behaviors offers a useful augmentation to the concept that slow-growing, long-lived and late maturing species are always the most vulnerable to become overfished. Productivity susceptibility analyses that rely heavily on life history traits have been used to identify at-risk species (*Hobday et al., 2011*; *Patrick et al., 2010*), but they have also received criticism for their inability to discriminate risk among species, except in the most extreme cases (*Hordyk & Carruthers, 2018*). Since a relationship between spawning behaviors and vulnerability has been observed (*Erisman et al., 2011*; *Sadovy De Mitcheson & Erisman, 2012*), and spawning behaviors are distinct from other life history traits, incorporating reproductive behaviors may improve the discriminatory power of vulnerability analyses. Despite the issues and limitations with discriminatory power, it is clear from the results of this study that the completion of a vulnerability analysis that includes spawning behavior remains a valuable management exercise to identify those species most vulnerable to fishing during spawning (*i.e.*, extreme cases). The results clearly define higher vulnerability and shows utility for prioritizing future research and improving monitoring efforts.

Vulnerability scores displayed increasing trends with species that exhibit large changes in relative abundance, have short spawning seasons and form transient aggregations, which is expected because those components are included in the calculation of intrinsic vulnerability. However, it is noteworthy that those trends persist because the overall vulnerability score includes extrinsic vulnerability factors, and the life history composite was more heavily weighted as an influence on vulnerability than any of the spawning

behaviors. Species with short spawning seasons that form transient spawning aggregations are prone to experience rapid depletion in response to targeted fishing pressure of spawners and slower recovery rates due to impacts on spatiotemporal egg production and stock recruitment relationships (*Claro et al., 2009*; *Erisman et al., 2012*; *Sadovy De Mitcheson & Erisman, 2012*). Yet, such declines can be difficult to detect using traditional, catch-based methods of estimating abundance or when historical information is unavailable (*Erisman et al., 2011*; *Maunder & Deriso, 2013*; *Lowerre-Barbieri et al., 2015*). Large increases in relative abundance during spawning are linked to enhanced catchability for most species that aggregate (*Ellis & Wang, 2007*; *Wilberg et al., 2009*). For those species, the increased catchability during spawning can also lead to hyperstability in which catch per unit effort (CPUE) remains high even while the actual abundance of the stock decreases in response to fishing mortality (*Erisman et al., 2011*). As a result, overfishing and stock declines may remain undetected until after sudden, large decreases in catch or CPUE occur (*Rose & Kulka, 1999*).

In contrast with a standard productivity susceptibility analysis (e.g., *Patrick et al., 2010*), our vulnerability analysis followed the approach of *Robinson & Samoilys (2013)* and *Robinson (2015)* by focusing specifically on identifying species that are highly sensitive to fishing during spawning. Extrinsic vulnerability was high for 25 of the 28 species, which reflects the high level of exposure to fishing during spawning throughout the GOM, and the existing threat of overfishing via the targeted, pervasive exploitation of spawning fish by commercial and recreational fishing activities (*Grüss et al., 2018*; *Grüss et al., 2019*; *Heyman et al., 2019*). The level of compliance with any of the regulations may vary and can greatly influence the effectiveness of the measure, but the regulations do serve as a metric of the perceived vulnerability by managers. Future research focused on assessing fishing effort, catch, and catchability (e.g., CPUE) patterns in relation to spawning as well as trends in market sales and values during spawning seasons would likely improve our understanding of vulnerability and the impacts of harvesting spawning fish on stock resilience for exploited fishes in the GOM (*Erisman et al., 2018*; *Heyman et al., 2019*). Understanding these behavioral impacts on resilience allows managers to better target management measures on spawning for those species most vulnerable to fishing due to such behaviors and discount those factors for species less vulnerable to spawning behaviors.

A detailed exploration of six important species further illustrates the management implications of our results. These species include examples of both coastal and reef fishes whose spawning behaviors fall along a continuum of intrinsic vulnerability scores (Table S4), ranging between resident (Spotted Seatrout, Red Snapper), mixed (Red Drum and Greater Amberjack), and transient aggregating species (Gag, Atlantic Goliath Grouper) (Table S3, Fig. 5). Red Drum and Spotted Seatrout are the two most important and valuable species for inshore recreational fisheries throughout the GOM (*Blanchet et al., 2001*; *National Marine Fisheries Service (NMFS), 2018*). Red Snapper (*Lutjanus campechanus*) is arguably the most important and politically contentious commercial and recreational fishery in the GOM (*Farmer, Froeschke & Records, 2020*). Both Greater Amberjack and Gag are also valuable, highly targeted species for both the recreational and

commercial sectors in the GOM that aggregate to spawn. Finally, Atlantic Goliath Grouper have been closed to commercial and recreational harvest in state and federal waters of the GOM since 1990 due to severe population declines associated with overfishing of their transient spawning aggregations (*Koenig, Coleman & Malinowski, 2020*).

Spotted Seatrout are classified as resident spawners, because they have a protracted spawning season (April–September; Table S3), and spawning occurs on a daily basis within small (*i.e.*, 10 s to a few hundred individuals) resident aggregations that persist at many locations and habitat types along the coast and throughout estuaries (*Saucier & Baltz, 1993*; *Walters et al., 2009*; *Biggs, Lowerre-Barbieri & Erisman, 2018*). Although Spotted Seatrout are managed at the state level and not assessed regionally, harvest levels have remained stable over the last 20 years in most areas (*National Marine Fisheries Service (NMFS), 2018*). Their resilience to persistent, intense fishing pressure is believed to be linked to their high level of spawning productivity, both spatially and temporally (*Biggs, Lowerre-Barbieri & Erisman, 2018*). For this reason, traditional catch controls such as daily bag and minimum size limits appear to be sufficient to maintain populations, and the targeted protection (e.g., seasonal restrictions or area closures) of spawning fish is not a priority for management.

Red snapper form schools of hundreds to thousands of fish throughout the year across a broad array of coastal and offshore habitats (e.g., natural and artificial reefs, petroleum platforms) in the GOM (*Patterson et al., 2007*; *Erisman et al., 2020*). At the population level, spawning occurs continuously from May through September in small groups within these localized schools rather than concentrating spawning at a few, highly populated sites (*Porch et al., 2015*; *Farmer et al., 2017*; *Glenn, Cowan & Powers, 2017*). While fishers have reported the previous existence of large spawning aggregation sites of Red snapper (*Lindeman et al., 2000*), there is no evidence that coordinated migrations to specific spawning sites occurs or that densities or relative abundance at spawning sites is markedly higher than non-spawning sites (*Patterson et al., 2007*; *Szedlmayer & Bortone, 2019*). Based on this information, we classified the species as resident aggregation spawners, but it would also be reasonable to classify the species as non-aggregating based on the specific criteria used (*Domeier, 2012*; *Claydon, Mccormick & Jones, 2014*). Regardless, the more important point is that the spatiotemporal dynamics of spawning behavior should confer reproductive resilience to intense fishing effort during the spawning season (*i.e.*, low overall vulnerability), which is supported by the results of this study and the status of the regional stock.

Fishing effort and landings for Red snapper in federal waters peak during the summer months in association with the recreational fishing season, which coincides with the peak spawning season for this species (*Heyman et al., 2019*). Nevertheless, rebuilding of the stock has occurred in response to strict catch and effort controls implemented at state and federal levels that did not focus on spawning (*SEDAR, 2018*; *Farmer, Froeschke & Records, 2020*).

Both Red Drum and Greater Amberjack (*Seriola dumerili*) are classified as mixed spawners (Table S3) with an intermediate level of intrinsic vulnerability to fishing in relation to spawning (Table S4). Red Drum are known to form large transient (hundreds to

tens of thousands of individuals) spawning aggregations that occur predictably at the mouths of estuaries and tidal inlets throughout the GOM from August to November, but these aggregations are not limited to discrete locations within such habitats, adults school year round, and resident aggregations have been reported as well (*Pearson, 1928*; *Holt, 2008*; *Lowerre-Barbieri, Burnsed & Bickford, 2016*; *Lowerre-Barbieri et al., 2019*). Although the fishery is closed in federal waters, it is open to recreational fishing in state waters throughout the GOM and commercially in state waters in Mississippi (*Heyman et al., 2019*). A rapid increase in the targeted harvesting of adult fish from their spawning aggregations by the commercial fishery during the mid-1980s led to overfishing and the closure of the federal fishery (*Porch, 2000*). However, recreational anglers continue to target these well-known spawning aggregations during the late summer and fall months and this activity peaks during the spawning season, but the impacts of this interaction on the population or the fishery have not been assessed. Similar to Red Drum, Greater Amberjack spawning behavior is classified as mixed with a spawning season from March to June in the GOM with regional variations in seasonality (Table S3). Commercial landings in federal waters are prohibited during the peak spawning season (March–May), while recreational landings for Greater Amberjack are closed during the end and after the spawning season (June–July) (*SEDAR, 2014a*). Recreational landings are consistently higher during the spawning season than the non-spawning season (*Kobara et al., 2017*; *Heyman et al., 2019*), but the impacts of fishing during spawning have not been evaluated.

Gag and Goliath Grouper are both transient spawners (Table S3) with a high intrinsic vulnerability to fishing during spawning in the GOM (Table S4). Gag exhibit complex reproductive ecology in which females form pre-spawning aggregations in coastal waters before migrating offshore to deep-water spawning sites in the winter where males tend to occur year-round (*Koenig et al., 1996*; *Carruthers et al., 2015*). The species is also protogynous, in which female to male sex change occurs during both the pre-spawning aggregations and at the spawning grounds (*Lowerre-Barbieri et al., 2020*). Unlike typical transient spawning aggregations that involve group spawning and hundreds to thousands of fish, Gag aggregations are comprised of tens to a few hundred fish and courtship occurs in pairs (*Gilmore & Jones, 1992*; *Coleman, Koenig & Collins, 1996*; *Lowerre-Barbieri et al., 2020*). GOM Gag were overfished and undergoing overfishing in the 1990s, with heavy fishing on Gag spawning aggregations that resulted in the severe reduction of males, which contributed to stock declines (*Coleman, Koenig & Collins, 1996*). Consequently, the protection of spawning Gag has long been a focal point for management and the rebuilding of the stock (*SEDAR, 2014b*; *SEDAR, 2016b*). Recent studies have demonstrated that Gag biomass and fishing pressure are highest in nearshore waters, pre-spawning aggregations are a spatiotemporal bottleneck to population productivity, and current regulations are not sufficient for the male population to recover to historic levels (*Carruthers et al., 2015*; *Lowerre-Barbieri et al., 2020*). Similar to Gag, Atlantic Goliath Grouper may migrate long distances (up to 500 km) to specific sites to spawn within transient spawning aggregations that form from late July through October (*Koenig et al., 2017*). Severe population declines occurred in the GOM from the 1950s through the 1980s due in part to overfishing of spawning aggregations, but the species has shown signs
of recovery following their complete protection from exploitation in state and federal waters in 1990 (*Koenig, Coleman & Kingon, 2011*).

One of the challenges to incorporating details of reproductive behavior within assessments and management is the paucity of information available. For example, discrete quantitative information on changes in the relative abundance (e.g., increases in fish density and abundance during formation of spawning aggregations) of fish within a given area in relation to spawning activity is rare and currently, has only published for red drum (*Lowerre-Barbieri et al., 2019*) and Gag (*Lowerre-Barbieri et al., 2020*). This represents a critical data gap relevant to the management of fish stocks in the GOM. Further, changes in relative abundance are directly tied to changes in catchability, which is widely accepted as a crucial component of vulnerability (*Arreguín-Sánchez, 1999*; *Salthaug & Aanes, 2003*; *Wilberg et al., 2009*) and as mentioned above, hyperstability. Our results also demonstrate that relative abundance is an important factor as it was not correlated to any life history traits. Therefore, it represents an independent aspect of reproductive behavior relevant to assessing vulnerability and should be the focus of further research for those species that form spawning aggregations that are targeted and heavily exploited by recreational or commercial fishing.

In contrast to changes in the relative abundance of fish during spawning, spawning seasons are well documented in the literature. However, spawning seasons are usually reported on a monthly scale, which does not capture the finer variations in spawning periodicity, the degree of aggregating behavior, or the distribution of spawning sites among species. For example, Cubera Snapper spawn from June through September, but aggregations only form at a few sites for 1–2 weeks each month, and actual spawning is restricted to just a few days within a single aggregation period (*Heyman et al., 2005*; *Biggs & Nemeth, 2016*). Nassau Grouper, Mutton Snapper, and other GOM species in the grouper-snapper complex exhibit similar behaviors in which spawning occurs over a few days within large aggregations that form at specific sites in synchrony with lunar (or semilunar) rhythms (*Heyman & Kjerfve, 2008*). On the other end of the behavioral spectrum, species such as Spotted Seatrout and Red Snapper can spawn daily (population level) over the course of 5 to 6 months, and spawning sites are numerous and widely distributed (*Walters et al., 2009*; *Lowerre-Barbieri et al., 2015*; *Glenn, Cowan & Powers, 2017*; *Biggs, Lowerre-Barbieri & Erisman, 2018*). Differences in the spatiotemporal dynamics of spawning affects their respective vulnerability, but much of that variation is captured in the categorization of aggregation type (e.g., Cubera Snapper form transient aggregations and Spotted Seatrout form resident aggregations).

There is also some potential variation with respect to aggregation type across the region. For example, studies on the reproductive behavior of Red Grouper conducted in the U.S. GOM have all concluded that this species does not migrate during spawning or form spawning aggregations. Red grouper exhibit a haremic mating system in which resident males pair spawn with individual females that reside in small groups within its home territory (*Coleman, Scanlon & Koenig, 2011*; *Nelson et al., 2011*; *Wall et al., 2011*). Conversely, studies of Red Grouper fisheries and populations off Campeche, Mexico in the southern GOM do refer to spawning aggregations based on inferences drawn
from increases in catchability and catch-per-unit effort during the spawning season at spawning sites (*López-Rocha et al., 2009*; *López-Rocha & Arreguín-Sánchez, 2013*). These discrepancies further demonstrate the importance of fisheries-independent research on spawning dynamics and behavior to inform fisheries management (e.g., areal or seasonal restrictions), including investigations of behavioral traits that may vary on regional scales within the GOM.

The benefit of the approach used in this study is that it can be used to inform state and federal management groups by identifying and prioritizing which species should be targeted for research, monitoring, and management actions in the absence of formal stock assessments. Of the species in our analysis that have not been assessed through formal stock assessments, Cubera Snapper, Warsaw Grouper, Sheepshead, Southern Flounder, and Black Drum are likely at risk of overfishing due to strong interactions between fishing and spawning. These species all exhibit reproductive behaviors that indicate they should be assessed in the GOM and that interactions between fishing and spawning should be investigated further to determine if spawning fish would benefit from additional protection (e.g., seasonal catch limits or inclusion with marine protected areas). Additionally, this approach can be applied to other regions and fisheries. As one example, the U.S. South Atlantic region contains many of the same harvested species as the GOM, but the stocks are managed separately and have different regulations in both state and federal waters. Additionally, the spawning seasons, locations, and environmental cues to spawning are well described for many stocks in the U.S. South Atlantic (*Farmer et al., 2017*). Therefore, our analysis can be transferred to that region with only minor adjustments to account for the differences in extrinsic vulnerability parameters. Our analysis also provides the framework to identify and prioritize management and monitoring of species that are vulnerable to fishing during spawning, which is especially important and applicable in areas where management resources, monitoring effort and fisheries data are limited (e.g., Caribbean, Tropical Eastern Pacific, Indo-Pacific) (*Salas et al., 2007*; *Gill et al., 2017*). Moreover, quantitative information on the periodicity and frequency of spawning and other spatiotemporal aspects of spawning behavior can be directly incorporated into estimates of annual reproductive output and spawning potential ratios (*Cooper et al., 2013*; *Erisman et al., 2014*, *2020*; *Lowerre-Barbieri et al., 2017*), thus representing a clear pathway to incorporate such valuable information within formal stock assessments.

## CONCLUSIONS

Exploited marine fish species displaying similar life history characteristics can be very different with regards to their vulnerability and resilience to fishing when reproductive behavior is considered. This distinction is meaningful, because spawning behaviors are underrepresented in the conservation or management plans of most marine fish species in the GOM and elsewhere. Based on the results of this study, it is important to consider both aspects within management frameworks, particularly for those species known to form large, transient spawning aggregations that are targeted by commercial and recreational fishing activities. Assessing the vulnerability of a marine fish species based on their size, longevity, and maturation rate alone may not capture the true complexity

of their biology and likewise their resilience or vulnerability to fishing pressure, particularly when fishing efforts target spawning sites, seasons, or the actual spawning period itself. As a result, incorporating spawning behavior within such analyses can significantly improve our understanding of vulnerability. By extension, the lack of biological and fisheries information on reproductive behavior hampers efforts to maintain healthy, productive stocks to benefit fisheries and ecosystems. Moreover, these types of bivariate frameworks can be a valuable tool for understanding the main factors underlying vulnerability of marine fishes and for prioritizing research and management around those species showing the greatest vulnerability to fishing during spawning.

## ACKNOWLEDGEMENTS

We are grateful to the many people that provided data, feedback, or other valuable support for this study including M. Karnauskas, W. Stearns, F. Parker, J. Lartigue, C. Young, A. Grüss, D. DeMaria, S. Hickman, K. Guindon, S. Cantrell, V. Ventura, W. Werner, D. Naar, S. Murawski, M. Russell, C. Taylor, T. Kellison, R. Williams, R. Crabtree, L. Crabtree, E. Reed, C. Koenig, K. Boswell, S. Fulton, J. Locascio, J. Brenner, B. Kirkpatrick, B. Gallaway, K. McCain, T. Wheatley, H. Binns, A. Acosta, A. Trotter, F. Giordano, F. Helies, R. Ellis, H. Staley, K. Faherty-Walia, R. Devictor, G.P. Schmall, B. Gorst, T. Loughran, L. Yeager, L. Fuiman, and many others.

Disclaimer: The scientific results and conclusions, as well as any views or opinions expressed herein, are those of the author(s) and do not necessarily reflect those of NOAA or the Department of Commerce.

### Funding

This work was funded by the NOAA RESTORE Science Program under award NA15NOS4510230 to the University of Texas at Austin. Additional Support was provided by an early career fellowship to BEE by the Gulf Research Program of the National Academies of Science, Engineering, and Medicine. The funders had no role in study design, data collection and analysis, decision to publish, or preparation of the manuscript.

### Grant Disclosures

The following grant information was disclosed by the authors:
NOAA RESTORE Science Program: NA15NOS4510230.
National Academies of Science, Engineering, and Medicine.

### Competing Interests

William Heyman is an employee of LGL Ecological Research Associates, Inc. Brad Erisman and Nicholas Farmer are employed by NOAA National Marine Fisheries Service. The authors declare that they have no other competing interests.

## Author Contributions

- Christopher R. Biggs conceived the study, compiled the data, designed the analytical framework, analyzed the data, prepared the figures and tables, wrote the paper, reviewed and edited drafts of the paper, and approved the final draft.
- William Heyman compiled the data, wrote the paper, reviewed and edited drafts of the paper, and approved the final draft.
- Nicholas A. Farmer compiled the data, reviewed and edited drafts of the paper, and approved the final draft.
- Derek G. Bolser compiled the data, analyzed the data, reviewed and edited drafts of the paper, and approved the final draft.
- Shinichi Kobara compiled the data, reviewed and edited drafts of the paper, and approved the final draft.
- Jan Robinson designed the analytical framework, reviewed and edited drafts of the paper, and approved the final draft.
- Susan K. Lowerre-Barbieri reviewed and edited drafts of the paper and approved the final draft.
- Brad E. Erisman led the project associated with this study as senior investigator, conceived of the study, compiled the data, designed the analytical framework, analyzed the data, wrote the paper, reviewed and edited drafts of the paper, and approved the final draft.

## Data Availability

The data compiled and included within the analyses are available in the Supplemental Files.

All data and information related to the study is also available at the project website at: https://geo.gcoos.org/restore.

## Supplemental Information

Supplemental information for this article can be found online at http://dx.doi.org/10.7717/peerj.11814#supplemental-information.

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
