# Peer review of "The importance of spawning behavior in understanding the vulnerability of exploited marine fishes in the U.S. Gulf of Mexico"

_PeerJ, doi:10.7717/peerj.11814_

## Round 0.1 · original submission · Major Revisions

This is a clever and well written paper on an interesting topic and as a result will make a good contribution to the literature. There are three very constructive reviews provided here which will greatly improve the manuscript.

In reading these reviews and the manuscript itself I note that:
The Discussion is too long
Figure 1 caption contains typos
Figure 3 (easier to understand) is a duplicate of Table 5 (more detail), so one needs to be removed.
Reviewers 1 and 2 both comment on species selection and the exclusion of species that don't aggregate. As such, a comment on this (or including such species in the analysis if it is justified) would be worthwhile.
Reviewer 2 comments on the weightings used in the vulnerability analysis. While I do not think this warrants rejection of the MS, a clear
and simple explanation about weightings (how the weightings were decided on and what the consequences of alternative weightings would be) would be useful.

Reviewer 1 ·

Basic reporting

In reviewing the current manuscript, I found the reporting to be fairly straightforward and generally clear. The Introduction, Methods and Results were all well-structured and clear to the reader. However, I found that some sections of the Discussion contained very long sentences that in many instances could be made clearer by spitting them into two sentences. I also found a few sentences missing words or having a verb with a wrong tense. These minor errors need to be amended to clarify meaning. The conclusions were otherwise sound. References were appropriate, relevant and complete. Tables and figures, as well as supplemental materials were well-conceived and the data within them clearly presented. All the figures were of high quality and the figure and table texts adequately described the content, however I do not feel it is appropriate to have the reader have to go to a separate website to find references tied to tables, i.e. Table 3. I also observed some line issues with Table 3 that may be the result of the upload, regardless these need to be amended.

Experimental design

The research clearly fits within the scope of PeerJ. The manuscript provides a much-needed focus on reproductive behavior in fisheries management, as the formation of spawning aggregations, that has been heretofore largely missing. The authors clearly state the need for this information and provide rigor in testing how reproductive behavior impacts fishing vulnerability. Although the general concept of the vulnerability of aggregating species to fishing has been long known, this manuscript provides the first (potential) peer-reviewed publication of its inclusion during a vulnerability analysis. The design of the study is based on the utilization of previously published data containing relevant life history parameters, however I did not find how multiple values for life history parameters were dealt with. As an example, some species may have had multiple published values of, for example, k or maximum age. The authors do not provide clarity in the methods for how these were dealt with or what values were used. If multiple values were used then a mean and SE would be appropriately reported. If a single published value was used, the authors need to explain why and how the decision to use this value was attained. Additionally, sexual pattern is widely known to have an impact on vulnerability, however I only found one instance of its mention in the discussion. Many of the fishes have published information on sexual pattern and it would be useful for the authors to explain to the reader why this life history parameter was excluded as it may have had an impact on the results. Finally, the study would have been improved and the argument made stronger by the inclusion of a few species that do not aggregate to spawn. I found this in some ways unfortunate in that it may have provided even more evidence of the need to consider aggregating behavior as a highly vulnerable life history trait. While the authors do provide a range of aggregating types among the chosen species, it would have been insightful to see how these species compare to non-aggregating species and how it might otherwise impact management. Otherwise, the methods were well-described and the data and methodology are generally sufficient to allow replication.

Validity of the findings

The authors make the impact and novelty of their study clear and unambiguous. All of their findings appear to be conclusive and where findings were not significant, the authors make the case clear as to why. The statistics are appropriate and the methodology is robust. I did not find anywhere where speculation was given, indeed the authors clearly show how reproductive behavior affects the species listed and stick strongly to the findings in their discussion and conclusions.

Additional comments

The manuscript provides a much-needed focus on reproductive behavior, as the formation of spawning aggregations, that has been heretofore largely missing from fisheries management decision-making. Although knowledge of aggregating behavior to increased vulnerability to fishing has been widely known for decades, to my knowledge this is the first study to examine this analytically. I, therefore, applaud the authors for taking on this exercise and highlighting the vulnerability of aggregating fish to overfishing. Although the manuscript is well-designed and generally well-written, I found a few grammatical errors that need amending. I also found several sentences, particularly in the discussion, that need to be split or otherwise altered to provide clarity. I have made comments throughout the manuscript where attention is needed. Although I did not find any major issues with the study, I nonetheless had some concerns regarding the exclusion of sexual pattern as a parameter within the study, as this has been shown to increase vulnerability within aggregating species. I would like the authors to respond as to why this life history trait was not included. I would also like to have the authors clarify how they dealt with multiple life history parameter values, e.g. k or size at sexual maturity. I make this request as I saw nowhere in the manuscript on how this was dealt with. If multiple values exist, Table 3 should report means and standard errors, or alternative if only one of multiple values was used, then why? I also note that there was no discussion of how these results compared with the Redlist designations for these species (where available) or the US Endangered Species designations. It would be fruitful to have this discussed. My only major concern is the use of terminology. I note throughout that the authors state that these species are vulnerable during spawning, however this is a deceptive term in many ways because the vulnerability is actually during and due to aggregating, not spawning. As the authors state, fish may aggregate for many days when spawning is not taking place, yet these species are vulnerable during these times. Indeed, a number of species don’t bite during actual spawning so the vulnerability is in actuality decreased. Therefore, throughout the manuscript the word ‘spawning’ needs to be replaced with ‘aggregating’. The authors can look at the comments within the pdf and get an idea of where this is needed. I would also comment that the study and the manuscript itself could have benefitted from the inclusion of a few non-aggregating species. Such inclusion would have given both readers and management some greater idea of where aggregating fish fit comparatively in the range of vulnerability. Otherwise, I would like to congratulate the authors on conducting this study.

Annotated reviews are not available for download in order to protect the identity of reviewers who chose to remain anonymous.

Reviewer 2 ·

Basic reporting

1. Raw data, writing style, references, and structure of this paper conform standards of the PeerJ journal.
2. However, I find that presentation of data requires some work: e.g., L326-341 could be shortened and listing of species names should be avoided. Some results are uninformative: e.g., L357-360.
3. I like the presentation of Figure 3, which demonstrates disassociation between spawning behavior and life-history traits clearly.
4. Discussion is too long—The scope of information on 6 species (L500-588) is beyond that of this paper. These parts should be shortened or removed.

5. Specific comments
a) L 318-321: This paragraph should be moved to Methods.
b) L321-325: Data on no. of references per trait seem unnecessary. Please remove these.
c) L363-364: The data do not match numbers in Table 5. Please double-check.
d) Table 1: Use consistent notation for max age, max weight, and max length: Amax, Wmax, and Lmax, respectively.
e) Table 6: I suggest to move this table to supplemental material as data of this table are presented in Figure 4.

Experimental design

1. Species selection: I thank authors for providing detailed procedures of species selection in supplemental material S1. They primarily selected species 1) those form spawning aggregations and 2) those are of conservation or fisheries importance. As a result of these selection processes, these data cannot be considered as random samples.

2. L250-252 and Figure 5: The non-random sampling may have impacted the analysis results in Figure 5. For example, the selected overfished stocks may contain primarily those associated with spawning aggregation. As such, the comparison of magnitudes of PC1 or PC2 between overfished and non-overfished stocks do not help to evaluate the additive explanatory power for spawning behavior traits.

To evaluate the additive explanatory power of spawning behavior traits, authors can conduct discriminant analysis or classification & regression tree with explanatory variables of 1) life-history traits vs. 2) life-history and spawning behavior traits.

3. Table 4: I am pleased to see data of lengths of spawning season for all study species. I am wondering if the results of PCA (Figure 4 and Table 6) may change if incorporating the no. of peak spawning months in this analysis.

4. Vulnerability analysis: I appreciate that authors provide detailed protocols of this assessment. I have two technical comments. First, as this assessment assumes equal weights between the two compound factors: extrinsic and intrinsic indicators, authors should provide reasons to justify this assumption. Second, with the a priori assignment of different weights for the individual intrinsic indicators, it disenables evaluating the species vulnerability with respect to these indicators. That is, this vulnerability analysis quantified stock vulnerability, but it could not evaluate vulnerability in association with spawning behavior. Related to the second comment I cannot find the association between vulnerability scores and overfishing statuses (L39-40) based on Figure 6.

Furthermore, the scope of vulnerability analysis seems to be much broader than the research question in this study. To make a coherent paper, I suggest to either to revise (by addressing the comments above) or exclude this vulnerability analysis.

Validity of the findings

The issues of 1) non-random sampling and 2) vulnerability analysis raised above may have impacted the results in Figures 4-6. In particular, the first issue makes it difficult to generalize the result. Thus, validity of findings cannot be justified at the present state.

Additional comments

Authors present a study that evaluates whether incorporating spawning behavior improves assessment of vulnerability of exploited marine fishes. Overall, this study provides abundant background information and high-quality of biological and management-related data for 28 marine fishes. I am especially pleased to see that authors apply published methods for quantifying spawning behavior traits among species.

However, some problems in the experimental design could have compromised the findings of this study. For the most, the procedures of species selection may have impacted the evaluation on discrimination of overfished statuses using the spawning behavior traits. Also, the design of vulnerability analysis disenable understanding the association between vulnerability and spawning behavior traits.

Although I appreciate authors’ work on data compilation, the species selection issues make it difficult to justify validity of findings. Consequently, I do not recommend to accept this paper. Nonetheless, I suggest that authors may revise or expand the vulnerability analysis on its own into a paper.

Reviewer 3 ·

Basic reporting

The structure of this paper conforms to PeerJ standards and guidelines. It is very clearly written with objectives/hypotheses plainly stated. Authors stick to these objectives throughout the manuscript and present relevant results and conclusions.
The introduction is well written and documented with relevant data and sufficient, valid references.
Tables 1-3, 5 and 6 are very clear and necessary. (However, I do not see a significant positive correlation between Aggregation type and Am or Lm in Table 5 as reported in text). Table 4 is not necessary since spawning months are listed in Table 3 for each species. Similarly, figures are in general clear and well described however Figure 3 may be superfluous as it appears to be a graphical version of Table 5. Although it shows a clear picture of the numerical table, I did not find it necessary. The authors may consider deleting these two. In addition, the caption in Figure 1 lists the weighted influence on vulnerability for intrinsic and extrinsic indicators. This part of the caption is not necessary and does not add to the graphical representation of the figure. These values are listed in Tables 1 and 2 where they make more sense.
Supplemental literature is supplied. Because this paper does not utilize original raw data but rather constructs indices using data analysis published in previous reports, no original raw data is required.

Experimental design

This paper uses a novel statistical treatment of data that has been collected for 28 recreational and commercially valuable fish in the Gulf of Mexico (GOM) by various researchers and fisheries programs. Initially the paper determines the relationship between life history parameters used in traditional stock assessment, and spawning behavior of these fishes. Once the lack of a relationship between the two is established using correlation coefficients, they use a treatment that addresses the vulnerability of the fish during spawning and compare it to the current status of each species. Although the methods used by the authors are standard statistical treatments, the addition of spawning behavior and extrinsic indicators into vulnerability models makes their approach unique, and arguably much more realistic and valuable as fisheries management tools. The simplicity of the model renders it potentially useful in places other than the GOM.

Validity of the findings

The Discussion and Conclusions are well stated, supporting and reflecting the Results. The examples of vulnerability for specific species used in the Discussion section were particularly helpful, illustrating the validity, practicality, and need of this new model which incorporates spawning behavior. The lack of spawning aggregation data, and in particular the relative abundance of fish at spawning sites, was highlighted as a gap in the vulnerability model. I found this to be overall an excellent and valuable paper.

---

## Round 0.2 · accepted · Accept

Thank you very much for the thorough response to the original reviewer comments. This level of attention to detail really helps the reviewers and myself understand the changes that have been made to address reviewer concerns. Well done and I look forward to seeing the final version of the manuscript.

Reviewer 2 ·

Basic reporting

Comments for Biggs et al. revision:

I am pleased to see that authors have improved description on species selection and the weightings of intrinsic and extrinsic factors, which help to increase clarity of the results. Also, coherence between the PCA and vulnerability analysis is now improved. Furthermore, discussion flows well. As a result, I have no further comments, and would be happy to recommend this paper to be accepted.

Experimental design

I have no further comments.

Validity of the findings

I have no further comments.

Additional comments

I have no further comments.